# Comparative Analysis of One Health Policies in Asia for Exploring Opportunities for British Columbia in Canada

**DOI:** 10.3390/ijerph22010034

**Published:** 2024-12-30

**Authors:** Benni Beltramo, Soumya Kolluru, Lisa Slager, Lindsey Wall, Kai Ostwald, Drona Rasali

**Affiliations:** 1School of Public Policy and Global Affairs, University of British Columbia, Vancouver, BC V6T 1Z2, Canada; benni.beltramo@gmail.com (B.B.); kolluru.soumya@gmail.com (S.K.); lindseyjean.wall@gmail.com (L.W.); kai.ostwald@ubc.ca (K.O.); 2School of Population and Public Health, University of British Columbia, Vancouver, BC V6T 1Z3, Canada; 3British Columbia Centre for Disease Control, Vancouver, BC V5Z 4R4, Canada

**Keywords:** One Health, public health, zoonotic diseases, ecohealth, climate change, policymaking, government

## Abstract

In response to emerging challenges that intersect humans, animals, and environments, there is growing international exigent need to adopt ‘One Health’ approaches. While One Health efforts are emerging in British Columbia in Canada, there are still challenges to overcome in the adoption of a One Health approach in policymaking. We conducted a comparative analysis of One Health policies in Asia, specifically, Singapore, Hong Kong, Bangladesh, and Thailand, which have well-established and sophisticated One Health approaches, to determine good practices in the implementation of One Health that could be considered for adoption in British Columbia. We conducted a literature review and scan of public-facing One Health websites, strategic action plans, and health databases, complemented by 13 semi-structured interviews with researchers, educators, service providers, human and animal health experts, and policymakers in our chosen Asian jurisdictions and British Columbia. While there was diversity in the One Health approaches taken by four jurisdictions, three key characteristics were present in policymaking processes in all of them: a national One Health strategic action plan, inter-ministerial coordination, and flexibility in the working relationships of public servants. One Health presents an opportunity for British Columbia to take a novel approach to public health policymaking, the one that is more holistic and effective at addressing shared health challenges.

## 1. Background

A global review highlighting the benefits of One Health approaches, with examples in joint health services, surveillance systems, antimicrobial resistance, food safety, environmental hazards, and zoonoses control, emphasized that these approaches are most effective for preventing, detecting, and investigating risks and hazards of endemic and neglected diseases [1]. It is estimated that 75% of emerging infectious diseases are zoonotic in nature (diseases that spread between animals and people) [2] infecting millions of people around the world every year [3]. There are wide ranging and heavy costs to these outbreaks: the COVID-19 pandemic resulted in nearly 6 million deaths by 2022 [4], spread over 200 countries, and was expected to generate a global economic toll of over USD 12.5 trillion by 2024 [5]. Scientific evidence of the origin of the mRNA virus (SARS-CoV-2) that causes the COVID-19 disease has most consistently indicated zoonotic origins of wildlife-to-human spillover routes through wildlife farming and wildlife trade, leading to the recommendation that the One Health approaches be incorporated into all jurisdictional public health, animal, and environmental health strategies [6]. During the pandemic, some increased efforts to operationalize a One Health approach at global, regional, and national levels ensued [7]. Thus, in response to emerging challenges that intersect humans, animals, and environments, there is growing international exigent need to adopt ‘One Health’ approaches.

Although there is no singular definition of One Health used around the world, the World Health Organization (WHO) defines it as an “an integrated, unifying approach that aims to sustainably balance and optimize the health of people, animals and ecosystems.”; the approach mobilizes multiple sectors, disciplines and communities at varying levels of society to work together to foster well-being and tackle threats to health and ecosystems, while addressing the collective need for clean water, energy and air, safe and nutritious food, taking action on climate change, and contributing to sustainable development [8]. In its most basic form, One Health places an emphasis on the interdependencies among human, animal, and environmental health (Figure 1). Recognizing that a single sector or siloed approach cannot solve complex and multifaceted issues, One Health is collaborative in nature and requires interdisciplinary inputs to detect, prepare, prevent, and respond to these challenges. Examples of issues currently being addressed by One Health include zoonotic diseases (e.g., COVID-19, ‘mad cow disease’, Avian influenza), antimicrobial resistance (e.g., salmonella), food safety and food security (e.g., sustainable agriculture), vector-borne diseases (e.g., West Nile, Lyme disease, malaria), emergency management (e.g., risk reduction and climate adaptation), environmental contamination (e.g., pollution, waste management), and mental health (i.e., our relationships with animals and the natural world) [9].

Traditionally, One Health has been utilized to address local and regional health challenges emerging from the human–animal–environment interface, such as outbreaks of Malaria, Ebola, and Avian influenza in regions with tropical climates. However, as a result of climate change, regions that have historically been more temperate are starting to see increases in average temperature, extreme weather patterns, and rates of zoonotic diseases [2]. For example, while British Columbia has not yet seen such high rates of these types of diseases compared to other regions in the world, within the last decade the province has still been forced to grapple with West Nile, ‘mad cow disease’, and the COVID-19 pandemic. These events have exposed the ways in which animal-to-human diseases can either emerge locally or transcend borders.

The importance of adopting a One Health approach has been made all the greater by climate change, which continues to shift the range of organisms we encounter and the emergence of diseases in places we may not have previously anticipated [10]. Viruses continue to evolve in response to human, animal, and environmental interactions. It is no longer a question of whether there will be another major global pandemic, but rather when it will occur, and whether we will be prepared for it.

### 1.1. One Health Context in BC

British Columbia, like any other Canadian province, provides publicly funded healthcare services under the system that is provincially budgeted and regionally delivered, with federal support to its public health and indigenous healthcare. The population in the province enjoys one of the best health outcomes, such as life expectancy at birth, a common health indicator, among the 13 provinces and territories in the country [11]. The concept of holistic health at the interconnection of people, animals, and lands are central to Indigenous perspectives, spirituality, and ways of being and have captured traditional knowledge since time immemorial across British Columbia. However, One Health efforts are only just beginning to emerge, like in many jurisdictions in the rest of the Western world, even though there are some leading representatives in genomics and clinical health research who are working closely with key government ministries on One Health frameworks and the economic feasibility of such an approach. Two COVID-19 outbreaks among mink farmers and their livestock in British Columbia detected symptoms and viral sequences from mink and human-derived isolates, pointing to a likely anthropogenic introduction of SARS-CoV-2 into farmed mink by farm staff, the viral evolution in the mink host, and then reintroduction into human hosts [12]. Detection of viruses from environmental samples underscored the importance of the environmental factors [13], clearly pointing to the need for a One Health approach to integrate the management of such outbreaks. There are also project-based One Health approaches to policy change. One such example is highlighted in the November 2021 announcement by the British Columbia government regarding its decision to permanently phase out the mink farming industry due to the “ongoing public-health risks associated with COVID-19” [14].

However, there are still challenges to overcome when adopting One Health policies in British Columbia, such as isolated project-specific efforts and difficulty scaling-up efforts to be formally institutionalized in government; difficulty in coordinating across sectors and agencies due to differing priorities, funding, resources, and data and information systems; a lack of shared knowledge across disciplines to assess and respond to risks, including the ways these risks affect humans, animals, and the environment; an absence of a uniform definition of One Health used across disciplines and sectors; and a lack of recognition or understanding of the importance of such an approach.

Despite these challenges, adopting One Health has immense benefits, including increased information sharing; interdisciplinary efforts in education, research, and policy; enhanced resilience to shared risks; and increased opportunities for promoting health across the One Health triad. This project sought to understand these challenges better and to look for ways to address them.

In order for British Columbia to better prepare for future global health emergencies and manage ongoing pandemic issues, the province needs to overcome existing challenges by implementing a One Health approach. One way to accomplish this is to find inspiration in ‘wise’ One Health practices among countries with existing One Health policies. Our project explores this pathway by offering a comparative analysis of One Health policies in Asia.

### 1.2. Why Asia?

Globally, Asia is home to most emerging infectious diseases (EIDs). The heat map of predicted relative risk distribution of zoonotic EID events showing the estimated risk of event locations with weighted model output reweighted by population (after factoring out reporting bias) accessed from https://www.nature.com/articles/s41467-017-00923-8 on 24 December 2024 [15] represented the probability of future zoonotic disease outbreak intensity across all countries [15]. Asia in general—and the Southeast Asian Region specifically—have witnessed multiple outbreaks of new viruses in recent history (e.g., Avian influenza, Swine influenza), resulting in functional and decade-long One Health policies in response to these health challenges as faced in the COVID-19 pandemic [16]. The innovative knowledge, actionable policy, and strong ethical principles of One Health can be learned from global collaborative networks, especially from the several practical experiences from Asia that demonstrate some of the One Health approaches for best practices such as the management of Nipah virus infections in Bangladesh and human liver fluke in Thailand [17]. Lessons and insights can be drawn from policies across Asian jurisdictions to strengthen British Columbia’s approach to One Health. Thus, for the scope of our project, four countries in the Asian region were appropriately selected for case comparison purposes.

## 2. Materials and Methods

Our project utilized both secondary and primary data. Secondary data was retrieved from a literature review and scans of public-facing One Health websites, strategic action plans, and health databases. This was complemented by 13 semi- structured interviews in order to fill in knowledge gaps and to provide more substantial, nuanced insights into the questions we posed.

Our team spoke, seeking officially available information with researchers, educators, service providers, human and animal health experts, and policymakers in British Columbia and our chosen Asia jurisdictions, Singapore, Hong Kong, Bangladesh, and Thailand. These countries demonstrated the highest level of sophistication in terms of One Health implementation for over a decade with formalized research, programs, and inter-ministerial collaboration for the preparedness, mitigation, and response to emerging diseases. Respecting the individuality of each interviewee, we recognized that everyone brought with them their own background, experience, and values to the issues at hand, as well as varying levels of experience with One Health policies.

By the nature of the interdisciplinary approach to One Health, there are many different contributors to this collaborative space. In questioning how research influences policy, our data was primarily shaped by the experiences and perspectives of those involved in research or academia. Many interviewees held multiple roles as researchers and front-line professionals or researchers and government workers. Ultimately, our analysis has been shaped by those traditionally perceived to be ‘experts’ by Western values that are embedded in education, knowledge, and institutional power structures. Outside the scope of this project exists the perspectives of traditional knowledge holders, public sector workers, community leaders, and private industry involved in One Health spaces. Although these interviews are not substantive enough to offer a robust and expansive look at all levels of One Health policy coordination, the diverse range of professionals and jurisdictions we spoke with provided us with unique insights to highly similar challenges.

## 3. Results and Discussion

Based on our desk review and interview interactions, four broad themes surrounding One Health policies have emerged from the analysis of the four case studies. These four themes include the ministerial governance and decision-making models for One Health policies, the avenues through which One Health research is conducted to inform policymaking, the funding modalities for One Health research and projects, and training initiatives for academics and professionals working in the One Health space.

### 3.1. Comparing One Health Models in Asia

A comparison of various One Health models in Bangladesh, Singapore, Thailand, and Hong Kong is shown in Figure 2. Each jurisdiction has a model through which One Health policies are formulated and implemented at the ministerial level. While key differences arise in the minutiae of each jurisdiction’s implementation framework, broadly speaking, there are similarities and shared characteristics among them. These frameworks can be grouped into three models: Coordinating Committee Model, Working Group Model, and Blended Model.

Supporting each jurisdiction’s One Health efforts is the existence of a national strategic action plan that encourages the use of a One Health approach to addressing challenges among human, animal, and environmental sectors. These documents are foundational; they promote cooperation across ministries, researchers, and professionals, and adopt values of cooperation, inclusivity, transparency, and active participation between actors. However, each jurisdiction’s implementation of its strategic action plan is highly dependent on its organizational model.

#### 3.1.1. Coordinating Committee Model

The primary characteristic of the Coordinating Committee Model is having a team of government senior-level officials, public workers, and researchers centralized within the national government structure. This entity provides top–down direction across multiple ministries to ensure that decision making is collaborative, coordinated, coherent, and in alignment with the government’s One Health strategic plan. In this model, an administrative arm provides technical and logistical support to the day-to-day functioning of the Coordinating Committee. This model represents how policy decisions are made in Bangladesh and Thailand.

In Bangladesh, the government’s One Health Coordination Committee encompasses three complementary groups [18]: The Inter-Ministerial Steering Committee, which includes senior decision-makers from the Ministry of Health and Family Welfare, Ministry of Fisheries and Livestock, Ministry of Environment and Forests, and Ministry of Agriculture, and is responsible for implementing Bangladesh’s Strategic Framework and Action Plan for One Health. The One Health Secretariat supports the logistics of inter-ministerial collaboration and conducts monitoring and assessment of the activities outlined in the strategic action plan. The One Health Technical Advisory Committee, comprised of leading researchers and experts, provides technical guidance on One Health policies.

In Thailand, the Coordinating Unit for One Health (CUOH) of the National Committee on Emerging Infectious Disease Preparedness and Response has been established. The CUOH is based within the Ministry of Public Health (MOPH) and is chaired by the Permanent Secretary for Public Health, while the Directors of the Bureau of Emerging Infectious Diseases, the Bureau of Epidemiology, the Bureau of General Communicable Diseases of the Department of Disease Control, and the MOPH serve as Co-Secretaries [19]. The CUOH serves as a focal point for One Health collaboration domestically and internationally and is guided by Thailand’s National Strategic Plan for Emerging Infectious Diseases. The CUOH Steering Committee supports these coordination efforts by providing technical and administrative guidance to operations [19].

#### 3.1.2. Working Group Model

Unlike the top–down approach seen with coordinating committees, the Working Group Model is a bottom–up organization in which a specialized and interdisciplinary team of experts at the clinical or university level provide advice and guidance to senior decision-makers across multiple ministries on health topics and policy issues.

In Hong Kong, there are two such working groups, both of which are housed within the Centre for Health Protection in the Department of Health, each focusing on a specific policy area: The High-Level Steering Committee on Antimicrobial Resistance (AMR) advises and coordinates governance and multi-sectoral policies regarding AMR. They oversee the implementation of Hong Kong’s Anti-Microbial Resistance Strategic Plan and ensure that implementation aligns with One Health principles [20]. The Scientific Committee on Zoonotic and Emerging Diseases advises on public health actions aimed at protecting the community from emerging diseases and provides the science behind the impacts of the changing social and economic conditions that play an integral role in the emergence of zoonotic diseases, such as population growth and mobility, food and eating habits, and shifting environmental factors [21].

#### 3.1.3. Blended Model

The Blended Model includes features of both the Coordinating Committee and Working Group models, offering a bi-directional flow of information to inform policy.

In Singapore, the minister-led One Health Coordinating Committee provides strategic direction, sets priorities on One Health issues, and champions interagency coordination across the whole of Singapore’s government [22]. The committee includes actors from the Ministry of Health, National Parks Board, National Environmental Agency, and Singapore Food Agency, who are responsible for overseeing the Anti-Microbial Resistance Strategic (AMR) Plan, which calls for the application of the One Health concept in AMR policies [23]. Under the direction of the One Health Coordinating Committee, the One Health Working Group formulates, coordinates, implements, and reviews health-related programs, initiatives, and policies. As needed, the One Health Coordinating Committee will also establish specific One Health project teams to advise on specific issues as they arise.

#### 3.1.4. Strengths and Weaknesses of Different One Health Models

The presence of a formalized coordination mechanism among ministries, such as that of a coordinating committee, signals legitimacy for collaborative approaches within government and creates a space where cooperative values can come to fruition. A coordinating committee allows ministers to co-create functional policies across departments, allowing them to overcome traditional challenges of siloed or disjointed approaches or contradictory policy efforts [24]. A strength of the coordinating committee model is the high level of coherence and cohesion on One Health actions throughout the government bureaucracy. However, such an approach is challenged by opposing priorities and agendas across ministries, and when visions or goals differ, it can slow decision-making processes.

Alternatively, working groups at the academic and research level tend to be more inclusive in their cooperative efforts. For example, Southeast Asia One Health University Network (SEAOHUN) expands beyond the human–animal–environmental health triad to collaborate with experts in economics, social sciences, urban design, etc. [25]. However, without collaboration at the government level, decision-making processes can be subject to ineffective policies across different departments, as working groups cannot guarantee coordination at the ministerial level. Working groups must also determine key leverage points and communication avenues to inform government decision-makers on emerging health concerns, which may not always be an effective or reliable way to inform leaders in emergency situations.

Singapore’s blended approach makes use of the strengths of the other two models, while simultaneously overcoming some of their weaknesses. A coordinating committee supports collaborative efforts across various ministries, while diverse working groups are assigned to address specific health challenges. There is a bidirectional flow of information, and working groups have a direct avenue to communicate and work with government decision-makers. That said, Singapore’s model is still challenged by varying institutional capacities, as the committee and working groups are currently looking to address inequalities across actors.

#### 3.1.5. Implications for British Columbia: Building a One Health Model

A provincial model for One Health in British Columbia is currently being envisioned. After speaking with leaders in this effort, we determined that the current model in British Columbia most closely resembles that of a Working Group Model, as seen in Hong Kong. Specifically, the research arm for the British Columbia Centre for Disease Control (BCCDC) has been proposed as the central hub for interdisciplinary knowledge sharing and programming across academia, researchers, health professionals, and government representatives. This follows a bottom–up approach to decision making, with an interdisciplinary team of researchers providing government decision-makers with science-directed and informed advice.

This type of model has been proposed for British Columbia because of existing institutional and governance structures. Government policy and programs can be directed by mandate letters and ministerial commitments. This allows for flexibility and lowers barriers for researchers to inform policy. In turn, many researchers and academics wish to communicate their findings to the government to inform policy. The government staff or official who receives this information depends on the applicable ministry for a particular issue, which becomes muddled when an issue spans across several government jurisdictions and falls under the responsibility of different decision-makers.

Importantly, there is an existing gap in the coordination mechanism at the government level in British Columbia, which makes it challenging to implement policies that cross jurisdictions. These barriers can be overcome by looking for inspiration in the Coordinating Committee and Blended Models found in Asia, which have demonstrated that the presence of a central unit comprised of key government officials and decision-makers across multiple ministries can signal government ‘buy-in’ and improve cooperation, coordination, and participatory decision making across multiple departments. Coordinating committee units also offer a direct line of communication between working groups and government decision-makers across all essential departments, while also allowing for regular information sharing and collaboration on shared challenges. Some form of top–down guidance stemming from government coordination across sectors exists in three of the four cases in Asia and is a common model adopted for One Health efforts. However, a Coordinating Committee does not currently exist in British Columbia, with different ministries being responsible for unique components of health (e.g., veterinary, environmental, public health, etc.).

### 3.2. One Health Research

#### 3.2.1. Conducting One Health Research in Asia

Research pertaining to One Health tends to take place in academic or clinical laboratories, which can exist in government research arms or outside of government in independent facilities, hospitals, or universities.

#### 3.2.2. Government Laboratories

Singapore has one of the more comprehensive government lab systems among the four jurisdictions in Asia [26]. The National Public Health Laboratory (NPHL) is a One Health research center that operates at the national level under the direction of the Ministry of Health, where knowledge is contributed by physicians and infectious disease experts (e.g., immunologists, epidemiologists, microbiologists). Further, under the National Parks Board are the Animal and Plant Health Center (APHC) and the Center for Animal and Veterinary Services (CAVS), both of which provide diagnostic services for surveillance and monitoring of human–animal diseases by leaders in veterinarian, wildlife, and environmental sciences. These government labs work in collaboration with a network of seven public hospitals and clinical labs across the country to advance One Health research [27].

Comparatively, in Bangladesh, One Health research takes place in the Laboratory of Food Safety and One Health. This lab exists within the larger International Center for Diarrhoeal Disease Research (ICDDR) and is considered a success in Bangladesh in part due to an expansive global research network, which includes contributions from renowned facilities like the Royal Veterinary College (UK), London School of Hygiene and Tropical Medicine (UK), Chatham House (UK), Sher-e-Bangla Agricultural University (Bangladesh), Bangladesh Agriculture University, and Massey University (New Zealand) [28]. Government-situated labs such as these allow health experts and academics to conduct tests and communicate findings that are essential for the monitoring, surveillance, identification, and response to infectious and communicable diseases. The results from research are directly shared with key government decision-makers, providing an immediate line of communication between research and policy. Further, research conducted in these labs supports surveillance efforts and improves the preparedness, mitigation, and response time of highly infectious pathogens. Bangladesh has adopted a four-way linkage of information between animal and human health epidemiology with animal and human laboratory information, which is then communicated to the One Health Secretary for quick decision making by senior researchers from these labs. This ensures the policy process is informed by the latest data and research [18,28].

That said, these labs are also challenged by their institutional reach. Unless labs have extensive networks that engage with those outside of government spaces—such as community leaders, front-line professionals, and industry—policy responses risk being ineffective on the ground.

#### 3.2.3. University/Clinical Laboratories

Although Hong Kong and Thailand do not have dedicated government laboratories for One Health, they do host sophisticated university research platforms. Hong Kong’s Center for Applied One Health Research & Policy Advice (CAOHRPA) is the key research center located at the City University of Hong Kong, with specialists working in infectious diseases, public health, aquatic animal health, veterinary sciences, applied research, and policy. CAOHRPA focuses on monitoring animal and human disease transmissions and food safety within supply chains and regularly contributes to publications by the Food and Agricultural Organization (FAO) [29].

Comparatively, Thailand’s research is facilitated by the Thailand One Health University Network (THOHUN), which operates under the Southeast Asia One Health University Network (SEAOHUN). THOHUN is a university network bringing together medical sciences, economics, pharmaceuticals, public health, veterinarian medicine, environmental sciences, dentistry, urban design, social sciences, agriculture, engineering, political sciences, law, and education from around the country [30]. Together, THOHUN provides One Health training to the future workforce by offering courses and workshops for early protection, prevention, and risk reduction of emerging diseases. THOHUN works in close collaboration with its primary funder, the United States Agency for International Development (USAID) [31]. The benefit of an expansive university network such as this means that a wider range of actors can be brought together, and the network is much more adaptive and flexible than those seen in government labs.

Research that is primarily conducted in independent clinical and academic laboratories, such as that of hospitals and universities, must share information with decision-makers in a different manner. For example, Hong Kong’s CAOHRPA does not directly inform government policy. Rather, it contributes to international publications and professional network groups that inform policy in academic spaces. THOHUN similarly informs policy through academic spaces, but also goes a step further by working with the media to perform One Health advocacy and educate the general public on important One Health communications.

#### 3.2.4. Implications for British Columbia: One Health Research and Information Sharing

One Health efforts in British Columbia occur in isolated projects or programs. Interviewees in British Columbia helped our team identify some of these efforts. Government bodies, such as BCCDC and PulseNet Canada, are currently leading in One Health initiatives across multiple ministries, departments, and jurisdictions. For example, BCCDC has pulled the Ministry of Health (BCMOH) and the Ministry of Agriculture, Food, and Fisheries (BCMAFF) into conversations regarding a One Health approach, and there is increased involvement from the Ministry of Environment and Climate Change Strategy (BCMECCS). The Ministry of Emergency Management and Climate Readiness (BCEMCR) plays a coordinating role across the province to address disaster and climate risks across four phases of emergency management (response, preparedness, mitigation, and recovery), including the development of a provincial risk assessment system that integrates disaster and climate risk, adopts the United Nations Sendai Framework on Disaster Risk Reduction, and establishes a holistic approach to risk reduction in collaboration with BCMECCS, the Ministry of Agriculture and Food (MAF), and other partner ministries. Further, the enactment of the new and modernized Emergency and Disaster Management Act in 2023 promises a new approach to managing disasters that reflects emerging understanding of interrelated risks.

As One Health grows in British Columbia, there is potential for further engagement with other important government bodies. There are also non-governmental stakeholders contributing to the One Health environment in British Columbia, including that of academic and clinical (public health, public policy, environmental sciences, veterinarians, etc.), non-profit (e.g., Genome BC), service organizations (e.g., First Nations Health Authority), and private (start-up ventures, industry) researchers. One Health research initiatives are conducted at universities and non-governmental research organizations. An opportunity for expansion of One Health research in BC would be to incorporate One Health principles into research initiatives taking place in government research facilities. BCCDC is currently working towards implementing such an initiative in the form of a One Health Genomics Research Centre.

However, experts we spoke with identified challenges in conducting multidisciplinary research and transitioning findings into policy. First, there is an increased emphasis on applied sciences, shifting away from research solely for academic purposes and instead advocating for and prioritizing research that has practical outcomes. Applied science has important implications for policymaking by actively seeking to inform key stakeholders and decision-makers to drive positive change with respect to human, animal, and environmental health. Applied research requires results to have real-world implications and for findings to be disseminated and communicated in order to inform policy change.

Secondly, there exists asymmetry and power imbalances among the various actors involved. Importantly, this means that the involvement of certain professionals across the One Health triad may not always be equal. This tendency was also present across Asia. In Hong Kong, One Health research is dominated by public health professionals, with veterinarians and animal health experts perceiving decision-makers and funders as being less receptive to their contributions. Further, the importance of environmental research, particularly that involving climate change, has traditionally been underrepresented in the One Health space. It is not only essential for British Columbia to increase representation across the One Health triad, but there also needs to be efforts towards inclusive and equitable engagement to ensure meaningful participation by all actors. Bangladesh has tried to address this challenge through rotational leadership among stakeholders working under its One Health Coordination Committee. Further, national strategic action plans across Asia lay out the values sought by the One Health approach, including actors to be involved and the mechanisms set in place to promote meaningful participation. British Columbia currently lacks an explicit One Health strategic action plan.

Lastly, interviewees stressed the importance of a concerted need for data sharing mechanisms in British Columbia, such as shared platforms or datasets to track incidence rates of emerging diseases in human and animal populations. Typically, a single team will be responsible for a given dataset. Sharing data depends on the context in which the data was collected and whether sharing information is approved by the team possessing the dataset, feasible under the specific platform the data is hosted on, or shareable under legal regimes (e.g., Access to Information Act). These are friction points in data sharing between teams and ministries within British Columbia’s government, and they restrict the ability for multiple teams to work together in the rapid, agile, and coordinated fashion required under One Health. The minute details of data sharing within the observed case studies were beyond the scope of this project; however, exploring this aspect of implementing One Health would be a valuable area for future research.

### 3.3. One Health Funding

#### 3.3.1. Funding Mechanisms for One Health in Asia

Funding for One Health stems from three main sources: government (bilateral, national, provincial, or local governments), non-government organizations (international, regional, local), and private (industry partnerships). Bangladesh, Singapore, Hong Kong, and Thailand all receive funding from multiple sources, yet their funding models may be dominated by one or two sources in particular, based on what is contextually relevant for each jurisdiction. Bangladesh receives significant funding from international organizations (IOs), specifically obtaining One Health project- and program-specific support from WHO, FAO, and UNICEF [32]. Comparatively, Singapore’s One Health initiatives are embedded in the national government structure, ensuring they have dedicated budget lines to fund their One Health programs and research labs [27]. One Health initiatives in Thailand and Hong Kong occur at the research and academic levels, where funding has been obtained from partnerships with bilateral government organizations (e.g., USCDC, USAID, WHO) [33,34]. Thailand also has emerging funding relationships with the private sector, such as obtaining financial support from Chevron for student scholarships and Pfizer to strengthen its AMR research available through South Asian One Health University Network [35]. Overall, international aid has played a key role in the funding of One Health across Asia.

Given that very few jurisdictions receive dedicated government funding (or if they do, the contributions are minor), interviewees throughout this project continually stressed their challenges in securing long-term, stable, and predictable funding for One Health. Project-specific grants or international aid have supported the establishment of new One Health programs and projects, but these grants often come with an end date that raises questions as to how these One Health systems will be left to fund operations in the medium and long term. Further, those in Hong Kong have identified trends in funding that prioritize medical and public health research over environmental and veterinary sciences, making it difficult for researchers to equally contribute to the One Health research space. To address some of these concerns with funding, Bangladesh has established a trust fund that consists of donations from supporters and One Health projects to finance its daily administrative costs and needs. As mentioned above, Thailand is looking to grow its private sector relationships and international networks to expand funding sources. Singapore has also identified international networks as potential sources of funding through the Association of Southeast Asian Nations (ASEAN) and the Global Water Research Coalition.

#### 3.3.2. Implications for British Columbia: Funding Opportunities

Funding models in British Columbia will be highly contextual, based on the availability of sources and the need. Funding One Health in British Columbia is still at a nascent stage, often determined on a project-to-project basis. This is partly because One Health has yet to be institutionalized at the government mandate level in British Columbia, where efforts can receive dedicated One Health budgets. Instead, early-stage financing in British Columbia’s One Health space is heavily characterized by the contribution of non-government actors. For example, Genome BC’s co-funding approach to research investment aided the launch of One Health pilot projects and research, such as the project that linked mink farming with the COVID-19 transmission risk [36]. Additional One Health pilot projects are planned for funding under Genome BC’s COVID-19 Rapid Response Funding Initiative [37]. This approach is unique as it is neither government-only funding nor an industry-only portfolio management approach, and thus relies on collaborative research agreements and commitments to milestones or outputs.

Interviewees raised concerns regarding the current funding models in British Columbia. For example, the Ministry of Health has one of the largest budgets in the province, yet would be contributing less for potential One Health efforts when compared with the Ministry of Agriculture and the Ministry of Environment. Exploring the potential opportunities of federal and provincial budget allocation in British Columbia is currently being undertaken by BCCDC’s business case for One Health. Therefore, this project only reiterates these funding concerns to complement current exploratory efforts occurring elsewhere. Further, the specifics of the economic feasibility of funding were a challenge throughout this project, as funding models in Asia were highly contextual and difficult to obtain information on in order to properly inform decision-makers in British Columbia. Potential future research could investigate what budget allocations would look like; the cost–benefit analysis of adopting preventative, mitigative, and holistic approaches; as well as other opportunities for funding, such as dedicated and shared One Health grants contributed by collaborative agreements with agencies, hospitals, and academic institutions, as well as expanding funding beyond academia into surveillance and reporting (an area that is suggested to be neglected in British Columbia).

The presence of non-profit actors in British Columbia’s funding model mirrors what is found in Asia, where non-profit organizations play an essential role in the establishment and ongoing operations of One Health projects, programs, and efforts across Bangladesh, Singapore, Hong Kong, and Thailand.

### 3.4. One Health Training

#### 3.4.1. One Health Training in Asia

Training and professional development programs were the most forward-looking aspects of One Health identified during this project, indicating an emphasis on developing a workforce in which the One Health concept is well socialized. The availability of education and skills training for those working with the One Health ecosystem was a key component identified by experts in Asia when it came to the effectiveness of One Health policy formulation and implementation. The following elucidates the different types of training programs that appeared in Bangladesh, Singapore, Hong Kong, and Thailand.

#### 3.4.2. Vocational and Professional Training

Singapore, Bangladesh, and Thailand offer vocational and professional training programs designated for those working within professional One Health spaces. This includes government officials, epidemiologists, and veterinarians. The benefit of these programs is that government officials and public workers are introduced and trained on key principles of One Health that support them in collaborative efforts across different jurisdictions.

#### 3.4.3. Field Training

In Singapore, Bangladesh, and Thailand, there is an emphasis on incorporating One Health principles in existing training and certification programs for epidemiologists through Field Epidemiology Training Programs. In each of these jurisdictions, field epidemiologists are considered the ‘front lines’ of One Health, as they are typically the personnel detecting and reporting outbreaks of zoonotic and emerging diseases [33]. Introductions to One Health prepare professionals for working with field professionals in other sectors and teach them to recognize key intersections between human, animal, and environmental health.

#### 3.4.4. Undergraduate Training

Hong Kong offers a One Health curriculum in its Veterinary and Life Sciences Undergraduate Program [38]. Similarly, a certificate program is offered at Singapore’s Nanyang Technological University in its School of Biological Sciences [39].

#### 3.4.5. Graduate Training

Singapore offers graduate-level training in One Health through a course at Duke Global Health Institute, which includes a focus on One Health [27]. Bangladesh’s One Health Institute at the Chattogram Veterinary & Animal Sciences University offers a master’s degree in Public Health (One Health) [40] and has collaborated with Massey University in New Zealand to offer a One Health Epidemiology Fellowship [41]. Both undergraduate and graduate training educates future professionals, academics, and researchers on the importance of One Health and introduces a more holistic perspective on the interdependencies among humans, animals, and their shared environment.

#### 3.4.6. Theme-Based Workshops and Seminars

All four countries host workshops and training seminars for existing professionals. For example, in Bangladesh, the One Health Hub offers seminars, workshops, and conferences for professionals, and in Thailand, there are opportunities for students and young professionals to participate in One Health workshops and training seminars through THOHUN [30]. One Health platforms, symposiums, or conferences are some of the rare opportunities in which a large range of stakeholders (from government to industry to academia to the public) can engage in conversation with one another and share information concerning One Health topics.

### 3.5. Policy Implications for British Columbia: Capacity Building and One Health Training

The existence of capacity building, knowledge sharing, and skills training opportunities across all four jurisdictions studied indicates the importance of educating future generations about the interdependencies of human, animal, and environmental health. Building knowledge, capacity, and confidence in this regard ensures policy formulation and decision making is further aligned with the core principles of One Health. Several interview respondents emphasized the role of training and education in their respective jurisdictions, thus further supporting this notion.

However, the existence of the above-mentioned instances is somewhat isolated since many jurisdictions around the globe do not offer One Health training or certification programs. This is also true for British Columbia. The presence of One Health education in British Columbia is difficult to detect and measure. There are existing informal efforts to share knowledge and build capacity. For example, in the Fall of 2021, BCCDC hosted a One Health Symposium where clinical researchers; academics; government officials; and human, animal, and environmental health experts met virtually to discuss emerging One Health efforts across Canada. However, these platforms are not institutionalized or formalized, as we have seen in Asia (e.g., Singapore’s One Health Platform hosted annual Symposiums prior to COVID-19 disruptions). Further, some academic instructors within health, animal, and environmental spaces are adding One Health concepts to their curriculum. For example, the University of Victoria offers a medical course (MEDS 487) on the interrelationship between human and animal health [42]. Other courses may adopt Indigenous concepts of holistic health that align with One Health values, despite not being framed as One Health. Looking beyond British Columbia, One Health efforts in Canada are also simultaneously undertaken in other provinces. The University of Guelph has designed a bachelor’s degree in One Health [43]. Despite there being isolated instances across British Columbia, the province does not currently offer formalized or targeted One Health training programs that provide certification or skills development for professionals working within the human–animal–environment interface, such as those present elsewhere.

Additionally, there is no university network connecting shared efforts in One Health. As demonstrated by THOHUN, establishment of a network can allow renowned university departments to provide their own areas of expertise. The University of British Columbia can lead in higher-level knowledge in population and public health, including global health (School Population and Public Health), global policy affairs (School of Public Policy and Global Affairs), and genomic and microbiological laboratory services and training (BCCDC). Meanwhile, the University of Fraser Valley may be better suited to provide hands-on training to One Health field practitioners.

### 3.6. Opportunities and Challenges

As One Health research often points to the need for combining tools and approaches from different areas of sciences [44], policies to advance One Health must integrate natural science, social science, and ethics to guide society’s plans for maintaining the health of the planet’s ecosystems [45]. When considering One Health implementation in British Columbia, it is crucial to highlight the opportunities and challenges that are currently present within the local context that can both enable and inhibit One Health policy efforts. By highlighting the current gaps and strengths, stakeholders are made more aware of drawbacks but also key opportunities to leverage for important policy change that has the potential to promote a One Health approach to decision making. Box 1 summarizes some of the enabling factors (A) and inhibiting factors (B) for consideration in One Health.

Box 1Enabling and Inhibiting factors of One Health in British Columbia, Canada.      A.ENABLING FACTORSOne Health research efforts are already emerging in Western science, with examples of BCCDC’s multidisciplinary working group and ongoing expert support in the creation of a One Health laboratory.The presence of resources and funding for One Health projects and information sharing within the non-profit space, such as efforts undertaken by Genome BC (e.g., mink pilot project, COVID-19 Rapid Response Funding Initiative) and BCCDC in offering expertise and guidance to provincial decision-makers.Informal programming in government structures allows for flexibility in policy and science-based decision-making in collaboration with knowledge holders, researchers, and academics.Existing efforts to assess the economic and political feasibility of a One Health approach in British Columbia.One Health symposiums for Canada-wide and regional conversations on One Health, such as the One Health symposium hosted by BCCDC.Efforts to incorporate One Health principals across world-renowned universities and institutions across British Columbia, such as those at the University of Victoria and Simon Fraser University incorporating One Health into their departments and curriculums.      B.INHIBITING FACTORSA lack of a formal coordination mechanism for multiple ministry actors and outside stakeholders operating in the One Health space to coordinate and share information. Further, key ministries like Education, are not actively involved in One Health development in the region.A lack of a national strategic action plan document that guides One Health efforts or calls for a One Health approach to decision-making and policy. This also contributes to a lack of understanding of One Health or the importance of such an approach across various levels of government and society.Few isolated instances of targeted One Health training programs, either for field professionals, government workers, or post-secondary students.Funding for academic projects without an applied research focus faces difficulty in transferring findings into practical outcomes, creating inefficiencies with funding. Currently, there are many projects in the academic research space, but comparatively fewer in the applied research space.A low level of integration among the numerous independent research projects being conducted in the region, including a lack of a formalized university network to connect isolated instances of One Health in academia, research, and clinical labs. This leads to inefficiencies in resource utilization.

## 4. Conclusions and Recommendations

As global health risks and climate emergencies continue to force our hand, the One Health approach is being implemented by governments around the world. Now is the time for British Columbia to overcome prevailing funding, logistical, capacity, and cooperative challenges and adopt a One Health approach that improves health outcomes for people, animals, and the environment. The benefits of a One Health approach are significant because One Health is multi-functional; it offers more than pandemic preparedness, as it encompasses food safety, food security, disaster risk reduction, climate adaptation, and agricultural sustainability to build long-term resilience to emergency events.

One Health presents an opportunity for the British Columbia government to approach public and environmental health differently than it has in the past. Understanding the importance of such an approach, there are existing efforts to adopt a One Health approach to policymaking in British Columbia, which this project aims to complement. The recommendations outlined in this report will support province-wide efforts in establishing a One Health framework to inform future health policy. For example, without coordinated efforts across multiple actors, the province risks maintaining the status quo of siloed decision making that is ineffective at addressing shared health challenges. Cooperating with non-government partners to implement these recommendations can catalyze change to overcome current policy challenges.

This project only represents the beginning stages of One Health efforts that were demonstrated in the Asian jurisdictions of Hong Kong, Bangladesh, Singapore, and Thailand and primarily focused on establishing systems and networks for cooperation, coordination, communication, and capacity building. There are further areas of interest that were beyond the scope of this project but are essential to investigate deeper to take full advantage of a One Health policy environment. When speaking with experts and leaders in the One Health space, concerns were raised on the lack of available platforms for data sharing as well as the legal and logistical constraints when transferring information between ministries and external stakeholders. Further, surveillance activities have some of the least integrated coordination between stakeholders, as well as the lowest allocations for funding, despite surveillance being one of the earliest forms of detection of potential health or environmental risks. However, inspiration in addressing these additional challenges can be found through further exploration of the approaches adopted in Asia, as well as in other regions leading in One Health, including Africa, Latin America, and the United States, and in local and Indigenous knowledge closer to home in British Columbia.

Advancing One Health in British Columbia will foster a value-based approach to risk management that is more holistic and effective at addressing shared health challenges, setting the stage for British Columbia to become a model for the rest of Canada. Taking on this challenge will advance British Columbia’s status as a leader in One Health.

## Figures and Tables

**Figure 1 ijerph-22-00034-f001:**
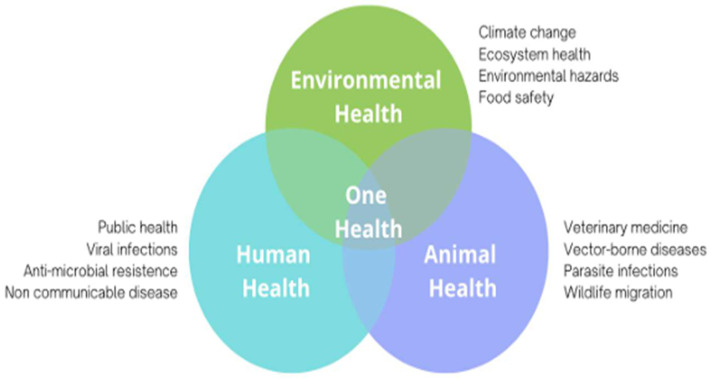
‘One Health’ at the intersection of animal, human, and environmental health.

**Figure 2 ijerph-22-00034-f002:**
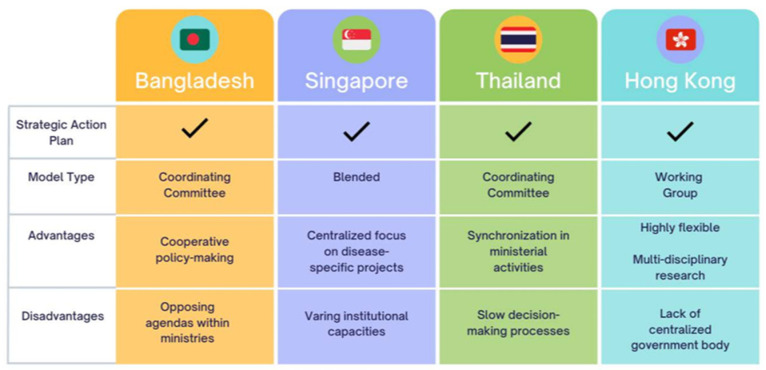
A comparison of One Health models in Bangladesh, Singapore, Thailand, and Hong Kong with their advantages and disadvantages.

## Data Availability

The details of aggregate data results from this study are presented in the included Appendix A. The raw data analyzed in this study are available on request from the faculty school through the corresponding author due to the privacy concerns as per the restrictions of the Research Review Board guidelines.

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
