# Peer review of "Comparative Analysis of One Health Policies in Asia for Exploring Opportunities for British Columbia in Canada"

_ijerph, 2024, doi:10.3390/ijerph22010034_

Round 1

Reviewer 1 Report

Comments and Suggestions for Authors

The manuscript submitted titled “Comparative Analysis of One Health Policies in Asia for Exploring Opportunities for British Columbia in Canada” aims to analyse and compare One Health policies from Asia (Singapore, Hong Kong, Bangladesh, and Thailand) and propose recommendations for British Columbia in Canada to adopt these approaches for enhanced public health preparedness.

This One Health approach is a new concept that represents an intersectoral response to emerging challenges that affect humans, animals, and the environment. The authors have emphasised the challenges to overcome when adopting One Health policies in British Columbia such as isolated project-specific efforts and difficulty scaling up efforts to be formally institutionalised in government; difficulty in coordinating across sectors and agencies due to differing priorities, funding, resources, and data and information systems; a lack of shared knowledge across disciplines to assess and respond to risks, in- including the ways these risks affect humans, animals, and the environment; an absence of a uniform definition of One Health used across disciplines and sectors; and a lack of recognition or understanding of the importance of such an approach. They have chosen to compare with Asian countries as they are prone to most emerging diseases, especially zoonotic diseases, and their experiences with the previous ones and lessons taken.

The subject is important regarding global public health, considering future zoonotic and other global pandemic risks. It is also novel in terms of the One Health approach.

1.     The introduction section provides sufficient background information with sufficiently recent references.

I can suggest adding some references dealing with One Health case studies at a global level [MINOR REVISION_1].

2.     Material and Methods section is well written. It details the literature sources and participant selection for interviews. No statistical analyses were performed as the study is qualitative in nature. But the methods are well-suited to the study. The target population consists of human populations, policymakers, researchers, and health professionals. Convenience sampling was used for selecting participants for interviews. A stratified sampling might be better representative.

3.     Results section: The results are clear and categorised into sub-headings like governance models, research mechanisms, and training initiatives.

No tables are present, nor are the figures related to the findings, the latter being used at the background information level.

I can suggest adding some figurative elements to summarise policy features besides zoonotic diseases [MINOR REVISION_2].

The strengths and weaknesses of Asian models are well described.

4.     Discussion section: Key findings are well discussed and supported by references with sufficient emphasis on the implications for British Columbia. A total of 34 references, ranging between 2011 and 2022 (mostly 2017 to 2022), were utilised, including global reports, academic studies and governmental papers.

5.     Conclusion section: They are supported by the findings of the study, and in concordance with the aim of the study, the applicability of Asian models to British Columbia was emphasised. Hence, the study achieved its aim of identifying policy recommendations for British Columbia.

6.     Limitations of the study: A narrow range of interview participants and the lack of a formal evaluation framework can be indicated as mentioned by the authors themselves.

To conclude, the study is of a local nature. But considering its type being “Case Report” and submitted for a special issue “One Health Including and Beyond Zoonoses”, my recommendation for this submission is “accepting with minor revisions.

Author Response

Top of Form

Review Report Form: Reviewer 1, Round 1

Reviewer 1 Round 1 Review Report:

Open Review

Quality of English Language

(x) The quality of English does not limit my understanding of the research.
( ) The English could be improved to more clearly express the research.

Yes

Can be improved

Must be improved

Not applicable

Does the introduction provide sufficient background and include all relevant references?

(x)

( )

( )

( )

Is the research design appropriate?

(x)

( )

( )

( )

Are the methods adequately described?

(x)

( )

( )

( )

Are the results clearly presented?

(x)

( )

( )

( )

Are the conclusions supported by the results?

(x)

( )

( )

( )

Comments and Suggestions for Authors

The manuscript submitted titled “Comparative Analysis of One Health Policies in Asia for Exploring Opportunities for British Columbia in Canada” aims to analyse and compare One Health policies from Asia (Singapore, Hong Kong, Bangladesh, and Thailand) and propose recommendations for British Columbia in Canada to adopt these approaches for enhanced public health preparedness.

This One Health approach is a new concept that represents an intersectoral response to emerging challenges that affect humans, animals, and the environment. The authors have emphasised the challenges to overcome when adopting One Health policies in British Columbia such as isolated project-specific efforts and difficulty scaling up efforts to be formally institutionalised in government; difficulty in coordinating across sectors and agencies due to differing priorities, funding, resources, and data and information systems; a lack of shared knowledge across disciplines to assess and respond to risks, in- including the ways these risks affect humans, animals, and the environment; an absence of a uniform definition of One Health used across disciplines and sectors; and a lack of recognition or understanding of the importance of such an approach. They have chosen to compare with Asian countries as they are prone to most emerging diseases, especially zoonotic diseases, and their experiences with the previous ones and lessons taken.

The subject is important regarding global public health, considering future zoonotic and other global pandemic risks. It is also novel in terms of the One Health approach.

  1. The introduction section provides sufficient background information with sufficiently recent references.

I can suggest adding some references dealing with One Health case studies at a global level [MINOR REVISION_1].

Authors’ response:

Additional more recent references have been added.

  1. Material and Methods section is well written. It details the literature sources and participant selection for interviews. No statistical analyses were performed as the study is qualitative in nature. But the methods are well-suited to the study. The target population consists of human populations, policymakers, researchers, and health professionals. Convenience sampling was used for selecting participants for interviews. A stratified sampling might be better representative.

Authors’ response:

Thank you for your advice. However, this being a Project report of case studies of selected promising countries in Asia to draw some learning for implications in British Columbia, a stratified sampling has not been feasible.

  1. Results section: The results are clear and categorised into sub-headings like governance models, research mechanisms, and training initiatives.

No tables are present, nor are the figures related to the findings, the latter being used at the background information level.

I can suggest adding some figurative elements to summarise policy features besides zoonotic diseases [MINOR REVISION_2].

Authors’ response:

A table (Table 1 Line 188-190) from the findings from the selected Asian countries has  been added now to summarize their policy features.

The strengths and weaknesses of Asian models are well described.

  1. Discussion section: Key findings are well discussed and supported by references with sufficient emphasis on the implications for British Columbia. A total of 34 references, ranging between 2011 and 2022 (mostly 2017 to 2022), were utilised, including global reports, academic studies and governmental papers.
  2. Conclusion section: They are supported by the findings of the study, and in concordance with the aim of the study, the applicability of Asian models to British Columbia was emphasised. Hence, the study achieved its aim of identifying policy recommendations for British Columbia.
  3. Limitations of the study: A narrow range of interview participants and the lack of a formal evaluation framework can be indicated as mentioned by the authors themselves.

To conclude, the study is of a local nature. But considering its type being “Case Report” and submitted for a special issue “One Health Including and Beyond Zoonoses”, my recommendation for this submission is “accepting with minor revisions.

Authors’ response:

Thank you for your valuable comments and recommendations. We have revised the manuscript accordingly.

Submission Date

15 November 2024

Date of this review

11 Dec 2024 02:35:54

Bottom of Form

© 1996-2024 MDPI (Basel, Switzerland) unless otherwise statedTop of Form

Reviewer 2 Report

Comments and Suggestions for Authors

The concept of One Health has been discussed for many years in conferences and in the literature. Post COVID-19, and with climate change causing subtle yet real changes in global disease transmission, the issue is even more important. The concept of One Health has become more important in recent years due to changes in interactions between human, animals, plants, and the environment. This needs to be stressed. For example, in the paragraph beginning with line 61, the authors should expound more on how global warming impacts vector-bourne diseases as well as food- and water-bourne diseases (e.g., changes in breeding areas, biting behavior, vector control). Lay the framework for the basis of One Health better by providing more of a background to the issue!

Since the authors talk about COVID-19 a lot, it should be noted that rarely animal coronaviruses can spread to people, yet it happened with SARS-CoV and MERS-CoV. A brief few sentences about COVID-19's origin should have been presented (e.g., Chinese researchers investigating the origin of the coronavirus outbreak in China have said that the endangered pangolin may be the "missing link" between bats and humans. Basically the authors could very briefly explain the animal model of COVID-19 to make their points. 

The following are more specific criticisms and recommendations for consideration:

I found the manuscript somewhat naive in scope and content and wish that the authors presented information about One Health in Canada (it exists!). I did a quick Google search and came up with a lot of background information that should be included in the Background Section. To whit, Global Affairs Canada and the International Development Research Centre have invested $40 million since 2021 to support One Health. Canada has received funding for One Health initiatives from many different sources, including Animal Health Surveillance System, One Health at the Community Level, Canadian Institutes of Health Research, etc etc.

Also, other regions of the world (e.g., Latin America) have embraced One Health and this should be mentioned. Latin America, for example, has embraced One Health initiatives for decades, especially in rural and underserved areas.

Line 35: the authors say that there is growing international pressure for One Health. By whom? Explain what you mean.

The paragraph beginning with line 61 is poorly written. Clarify what you mean to say. To what extent have other Canadian provinces embraced/adopted a One Health policy? Would such programs be an appropriate prototype for this study?

The write up on the Asian case studies is a bit long, but does provide interesting information. To what extent these examples can translate into policy in BC remains to be seen. What may "work" in one country may not in another. 

On a somewhat more petty note, update your references!

Author Response

Top of Form

Top of Form

Review Report Form: Reviewer 2, Round 1

Open Review

Quality of English Language

(x) The quality of English does not limit my understanding of the research.
( ) The English could be improved to more clearly express the research.

Yes

Can be improved

Must be improved

Not applicable

Does the introduction provide sufficient background and include all relevant references?

( )

(x)

( )

( )

Is the research design appropriate?

( )

(x)

( )

( )

Are the methods adequately described?

( )

( )

( )

( )

Are the results clearly presented?

( )

(x)

( )

( )

Are the conclusions supported by the results?

( )

( )

( )

(x)

Authors’ response:

Thank you much for your comments. Introduction and research contextualization and presentation of results have been majory revised and greatly improved.

Comments and Suggestions for Authors

The concept of One Health has been discussed for many years in conferences and in the literature. Post COVID-19, and with climate change causing subtle yet real changes in global disease transmission, the issue is even more important. The concept of One Health has become more important in recent years due to changes in interactions between human, animals, plants, and the environment. This needs to be stressed. For example, in the paragraph beginning with line 61, the authors should expound more on how global warming impacts vector-bourne diseases as well as food- and water-bourne diseases (e.g., changes in breeding areas, biting behavior, vector control). Lay the framework for the basis of One Health better by providing more of a background to the issue!

Authors’ response:

The concept of One Health has been emphasized throughout the manuscript. In the revised version of our manuscript, this has been done by adding one sentence based on a global review in the first sentence,

Since the authors talk about COVID-19 a lot, it should be noted that rarely animal coronaviruses can spread to people, yet it happened with SARS-CoV and MERS-CoV. A brief few sentences about COVID-19's origin should have been presented (e.g., Chinese researchers investigating the origin of the coronavirus outbreak in China have said that the endangered pangolin may be the "missing link" between bats and humans. Basically the authors could very briefly explain the animal model of COVID-19 to make their points. 

Authors’ response:

While COVID-19 and its origin has been described, discussed and debated in numerous articles published in the scientific literature thought the period of pandemic, we have only added the information of a case of SARS-Cov-2 virus in minks-human-environment interactions in British Columbia, which is directly relevant to the interest of British Columbia (Line 87-94).  

The following are more specific criticisms and recommendations for consideration:

I found the manuscript somewhat naive in scope and content and wish that the authors presented information about One Health in Canada (it exists!). I did a quick Google search and came up with a lot of background information that should be included in the Background Section. To whit, Global Affairs Canada and the International Development Research Centre have invested $40 million since 2021 to support One Health. Canada has received funding for One Health initiatives from many different sources, including Animal Health Surveillance System, One Health at the Community Level, Canadian Institutes of Health Research, etc etc.

Authors’ response:

Of course, Canada has invested significantly to One Health, but most of this goes out of the country through Global Affairs Canada’s international development program. Unfortunately, because health is primarily a provincial mandate within Canada, and Provinces like BC, with more that 40 per cent of provincial budget going to health sector, are still focusing on acute health care, preventive health, especially One Health is under the shadow without a solid strategy. Our paper is out of scope for examining all of the priorities of health sector. Currently, we are only examining what might be learning from selected Asian countries which have more prominent One Health policy.  

Also, other regions of the world (e.g., Latin America) have embraced One Health and this should be mentioned. Latin America, for example, has embraced One Health initiatives for decades, especially in rural and underserved areas.

Authors’ response:

As mentioned above, currently, we are only examining- as a starting point -what might be learning from selected Asian countries which have more prominent One Health policy. 

Line 35: the authors say that there is growing international pressure for One Health. By whom? Explain what you mean.

Authors’ response:

There is no documented direct pressure, but it is felt through the global, regional and national scenarios and happenings. Now, it has been clarified by adding a literarture information (Line 38-45). We have now changed the word “pressure” to “demand” which sounds more appropriate after the reviewer’s comment.

The paragraph beginning with line 61 is poorly written. Clarify what you mean to say. To what extent have other Canadian provinces embraced/adopted a One Health policy? Would such programs be an appropriate prototype for this study?

The write up on the Asian case studies is a bit long, but does provide interesting information. To what extent these examples can translate into policy in BC remains to be seen. What may "work" in one country may not in another. 

Authors’ response:

Thank you for your valuable comment. As this is a specific project report on a specific Asian case studies, they have been narrated in a bit lengthy description, which we believe is appropriate.

On a somewhat more petty note, update your references!

Authors’ response:

All the references have been checked, and updated with additional ones as well.

Submission Date

15 November 2024

Date of this review

01 Dec 2024 23:10:45

Bottom of Form

© 1996-2024 MDPI (Basel, Switzerland) unless otherwise stated

Round 2

Reviewer 2 Report

Comments and Suggestions for Authors

The authors have edited their piece, which makes the manuscript stronger, in my opinion. I particularly like Box 1, which nicely summarizes things.

I would ask that the authors seriously think about including a few sentences describing the Canadian health care system (provincial-based) and explain why they are focusing only on BC (aside from the fact that they live and work in this province). This will make the study seem less random--why BC?? What is the rationale for selecting this province. 

The inclusion of the word "demand" is too harsh. Demand is a strong word and there is no evidence that I could find of a "demand" to adopt One Health. Why not say "desire" "concern" or rework the sentence to use "exigent". 

Author Response

Review Report Form Round 2

Open Review

Quality of English Language

(x) The quality of English does not limit my understanding of the research.
( ) The English could be improved to more clearly express the research.

Yes

Can be improved

Must be improved

Not applicable

Does the introduction provide sufficient background and include all relevant references?

(x)

( )

( )

( )

Is the research design appropriate?

(x)

( )

( )

( )

Are the methods adequately described?

(x)

( )

( )

( )

Are the results clearly presented?

(x)

( )

( )

( )

Are the conclusions supported by the results?

(x)

( )

( )

( )

Comments and Suggestions for Authors

The authors have edited their piece, which makes the manuscript stronger, in my opinion. I particularly like Box 1, which nicely summarizes things.

Authors’ response: Thank you much for your positive and encouraging comment. Much appreciated.

I would ask that the authors seriously think about including a few sentences describing the Canadian health care system (provincial-based) and explain why they are focusing only on BC (aside from the fact that they live and work in this province). This will make the study seem less random--why BC?? What is the rationale for selecting this province. 

Authors’ response:  As advised, we have added the context and rationale for selecting BC province in a coupled of sentences in the beginning as given below for revising the section One Health Context in BC: “British Columbia like any other Canadian provinces provides publicly funded health care services under the system that is provincially budgeted and regionally delivered, with federal support to its public health and indigenous healthcare. The population in the province enjoys one of the best health outcomes, such as life expectancy at birth, a common health indicator, among the 13 provinces and territories in the country.”

The inclusion of the word "demand" is too harsh. Demand is a strong word and there is no evidence that I could find of a "demand" to adopt One Health. Why not say "desire" "concern" or rework the sentence to use "exigent". 

Authors’ response: Thank you much for your suggestion. We have changed the word, “demand” to “exigent need” which is sounds greatly appropriate in the context we have been trying to reflect.

Submission Date: 15 November 2024

Date of this review: 22 Dec 2024 19:04:49